# HyperCLIP: Adapting Vision-Language models with Hypernetworks

## Abstract

Self-supervised vision-language models trained with contrastive objectives form the basis of current state-of-the-art methods in AI vision tasks. The success of these models is a direct consequence of the huge web-scale datasets used to train them, but they require correspondingly large vision components to properly learn powerful and general representations from such a broad data domain. This poses a challenge for deploying large vision-language models, especially in resource-constrained environments. To address this, we propose an alternate vision-language architecture, called HyperCLIP, that uses a small image encoder along with a hypernetwork that dynamically adapts image encoder weights to each new set of text inputs. All three components of the model (hypernetwork, image encoder, and text encoder) are pre-trained jointly end-to-end, and with a trained HyperCLIP model, we can generate new zero-shot deployment-friendly image classifiers for any task with a single forward pass through the text encoder and hypernetwork. HyperCLIP increases the zero-shot accuracy of SigLIP trained models with small image encoders by up to 3% on ImageNet and 5% on CIFAR-100 with minimal training throughput overhead.

## 1 Introduction

A now-standard approach in deep learning for vision tasks is to first pre-train a model on web-scale data and then adapt this model for a specific task using little or no additional data. Despite the widespread success of these models and their lack of reliance on large-scale labeled datasets, a significant downside is that these models are often on the order of billions of parameters – much larger than their supervised counterparts for a given task at the same accuracy level.

While these pre-trained models are powerful due to their generality, practitioners still need to apply them to well defined and specific tasks. We consider settings where there are additional constraints on the size of these models such as in edge computing applications. Within this context, strategies to reduce the memory footprint or inference latency of these massive models are of paramount importance.

There exist a variety of such strategies broadly categorized into pruning, quantization, and distillation methods (Sun et al., 2023a; Dettmers et al., 2022; Frantar & Alistarh, 2023). These methods often involve first training a large model and then applying the chosen technique in a post-hoc fashion. Additionally, many of these methods require specialized hardware support to achieve actual memory and latency reductions (Liang et al., 2021; Yu et al., 2017; Han et al., 2016).

We propose a method of pre-training vision-language models (VLMs) that allows us to derive small vision models appropriate for deployment on edge devices without requiring multi-step training procedures or any specialized hardware. We suggest a new contrastive learning architectural design based on hypernetworks that improves performance over current state-of-the-art baselines. Our architecture can additionally be used in conjunction with a variety of model compression methods for further memory or latency improvements.

The enormous size of image encoders in VLMs is a direct consequence of the scale of their pre-training datasets: the model's image encoder is tasked with learning representations across an extraordinarily large data domain, and we show that small vision encoders struggle to learn such a breadth of representations. In this work, we propose a new strategy. Instead of fixing one image

Figure 1: (Left) The traditional CLIP architecture with SigLIP loss. (Right) the HyperCLIP variant. Overview of HyperCLIP. We use an hypernetwork to generate the weights of a smaller vision model within the SigLIP contrastive pre-training framework. The entire setup is trained end-end. HyperCLIP increases the zero-shot accuracy of SigLIP models with small image encoders by up to 3% on ImageNet and 5% on CIFAR-100 with minimal training throughput overhead.

encoder that needs to account for all possible image captions, we instead adaptively precondition the image encoder based on each particular text input. By setting the weights of the image encoder specifically based upon a given text embedding, we are able to use much smaller image encoder vision networks that are automatically specialized to each task.

We accomplish this goal through the use of a hypernetwork. Our hypernetwork takes in one or more embeddings from our VLM's text encoder and outputs the weights of a subset of the image encoder model. In this way, the hypernetwork learns the model weights necessary to represent an image as a function of text associated to that image. This hypernetwork is trained jointly with the usual text and image encoders present in VLMs, and is compatible with any type of contrastive pre-training. Our method, which we call **HyperCLIP**, allows for the usage of much smaller image encoders, resulting in inherent compression, i.e. fewer model parameters and faster inference in the deployed model.

We find that the performance of small vision models can be improved by several percentage points across a range of tasks when their weights are adapted via HyperCLIP. We show that using HyperCLIP to adapt *only the normalization layers* of several widely used small vision models is sufficient to improve their performance on standard zero-shot classification benchmarks. In some cases, we even find that a hypernetwork-adapted small vision model is able to outperform a larger non-adapted vision model.

## 2 BACKGROUND

**Preliminaries**: We formalize the contrastive pre-training setup, the use of hypernetworks, and the associated notations.

**Contrastive vision-language pre-training** models like CLIP (Radford et al., 2021) and its successor SigLIP (Zhai et al., 2023) learn powerful vision representations by simultaneously training an image encoder and text encoder on a large dataset of images and associated text. Informally, the objective is to bring positive pairs (containing relevant image-text pairs) closer and push negative pairs (containing unrelated image-text pairs) apart in the learned embedding space. Let the image encoder and text encoder be denoted as $\mathcal{F} \colon \mathbb{R}^{B \times I} \to \mathbb{R}^{B \times D}$ and $\mathcal{G} \colon \mathbb{R}^{B \times C} \to \mathbb{R}^{B \times D}$, respectively, where $B$ is the batch size, $I$ is the dimensionality of the image input, $C$ is the contextual input dimension of the text (e.g., number of tokens or sequence length), and $D$ is the embedding dimension of the output. Further, let the parameters of the image encoder be denoted by $\Theta = \{\theta_1, \ldots, \theta_L\}$, where $\theta_l$ are the parameters of layer $l$, and $L$ is the total number of layers.

As a natural consequence of the constrastive training objective, a trained CLIP or SigLIP model may be trivially repurposed as a $K$-class image classifier: given an image embedding $x \in \mathbb{R}^D$ and a set of text embeddings $Y \in \mathbb{R}^{K \times D}$ each representing a candidate class, the matrix-vector product

$Yx \in \mathbb{R}^K$ represents similarities between the image and each class, and $\arg\max\limits_{i \in \{1,...,K\}} Yx$ will select the highest similarity text embedding, constituting the model's class prediction.

SigLIP, a state-of-the-art improvement to CLIP, uses a sigmoid loss for the contrastive loss objective and can be more memory efficient because each pair is treated as an independent term in the loss, allowing the loss computation to be distributed across devices with a reduced memory footprint.

Let $Z$ be a matrix with 1's on the diagonal and -1's elsewhere, which serves as the labels for the image-text pairs. We can then define the siglip objective optimized over a mini-batch:

$$\mathcal{L}_{\text{SigLIP}}(X, Y) = -\frac{1}{B} \sum_{i=1}^{B} \sum_{j=1}^{B} \log\left(\sigma\left(Z_{ij}(\eta(X_i \cdot Y_j) + \zeta)\right)\right) \tag{1}$$

where $\eta, \zeta \in \mathbb{R}$ are learnable parameters and $\sigma$ is the sigmoid function.

**Hypernetworks** (or *hypernets*) are trained to produce the weights of another neural network (which we call the *mainnet*) given some input. There are no fundamental constraints on the architecture of a hypernet and they may be trained to produce the entire weight or a subset of the mainnet. The idea of hypernetworks trained end-end date back to fast weights Schmidhuber (1992); Gomez & Schmidhuber (2005) which are networks trained to produce context dependent weight updates for a second network. Hypernetworks have also been designed to work with a range of modern architectures including ConvNets, RNNs, and transformers, among others (Denil et al., 2013; Bertinetto et al., 2016; Jia et al., 2016; Ha et al., 2016), and are particularly well used in Bayesian formulations to several applications of deep learning where the hypernetwork learns to generate distributions over the weights of the mainnet.

Formally, let a hypernet be denoted as $\mathcal{H}(X_h; \Phi)$, and the mainnet be denoted as $\mathcal{M}(X_m; \Theta)$, where $X_h$ and $X_m$ are the inputs to the hypernet and mainnet respectively, and $\Phi$ and $\Theta$ are their respective weights. In a typical setup, $\mathcal{H}$ is designed to output the entire set of mainnet parameters $\Theta$, and any loss used is back-propagated through the hypernet weights $\Phi$. The hypernet, $\mathcal{H}(X_h; \Phi)$ may function as a dynamic parameter generator for the mainnet, $\mathcal{M}(X_m; \Theta)$. In such a scenario, the hypernet takes a set of conditioning inputs $X_h$, which could represent various contextual information or meta-parameters, and generates the weights $\Theta$ for the mainnet based on those inputs. This allows the mainnet to adapt its parameters dynamically to different tasks or data distributions without the need for extensive retraining. However, hypernetworks have been shown to be notoriously difficult to train in practice, largely because there are no principled ways to initialize them, given that their outputs directly influence the optimization landscape of the entire training process (Chang et al., 2023; Beck et al., 2023).

## 3 HYPERCLIP: A METHOD FOR SMALL-SCALE IMAGE ENCODERS IN VISION-LANGUAGE MODELS

In this section, we present the HyperCLIP architecture, a new approach to pre-training vision-language models (VLMs such as CLIP), which allows for a *much* smaller image encoder, able to eventually be deployed on edge devices. Naturally, compressing the entire knowledge of a VLM into such a small network is challenging: typical VLMs have a very large number of parameters in both the text and image encoders, and indeed derive much of their performance boost from this scale.

To motivate the development of our HyperCLIP network, we consider a common "zero-shot" or "one-shot" application of CLIP as an image classifier. In this use case, one takes a common image classification task (say, classifying images into one of ten categories), develops a set of text prompts for each category, and constructs the embeddings for each prompt using the CLIP text encoder. When we want to classify a new image using this system, we then embed the image using the CLIP image encoder, find which prompt is closest in embedding space, and output the corresponding class. These embeddings can further be fine-tuned (if desired) using a small amount of labeled training data (the so-called "few shot" setting). This pipeline was originally proposed in Radford et al. (2021), and is illustrated in Figure 1.

Now, consider the setting where we want to use this same process, but deploy the eventual classifier onto a small embedded device. A direct application would be challenging, owing to the fact that

the image encoder for CLIP is still quite large. However, for the most part, such a large encoder would not be strictly required for the final deployment of the classifier: only a "small part" of CLIP's internal knowledge is actually needed for a given image classification problem, and thus it should be possible to distill a smaller classifier for a particular classification problem. However, traditional distillation is quite inefficient in that it requires a existing set of training data, and it is unclear how to best leverage the full CLIP (i.e., teacher) model for this task since it does not directly produce class probabilities.

The basic intuition of the HyperCLIP model is that we can directly train a VLM that skips this explicit distillation process entirely, and instead produces an image encoder that is *already* optimized for use on a particular classification problem. In order to achieve this, we leverage a hypernetwork that produces a specialized image encoder directly for some subset of textual prompts. In the remainder of this section, we describe our HyperCLIP model in detail and highlight the different design choices.

### 3.1 THE HYPERCLIP MODEL

At a high level, our HyperCLIP model consists of three main components, two of which are direct analogues of the traditional CLIP model, and one of which is a new component: 1) the image encoder, which takes images and produces vectors in the CLIP embedding space, 2) the text encoder, which takes text input and produces vectors in this same embedding space; and 3) the new component of a hypernetwork, which maps text embeddings to certain parameters of the image encoder itself. The first two components are essentially identical to that of the traditional CLIP model: the text encoder is in fact directly taken from CLIP, and the image encoder has the same functional form as the CLIP image encoder, except that the network is substantially smaller, given that we want to run it on edge devices. This poses an obvious challenge, however, to learning a suitably expressive encoder as in the traditional CLIP model.

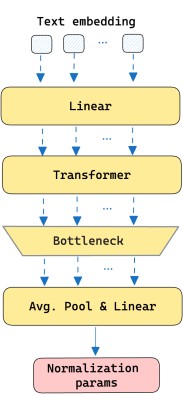

Figure 2: (Left) Overview of the hypernetwork. We process the text embedding using a transformer and directly output the normalization scale and bias parameters.

To address this problem, we propose the new element of the Hyper-CLIP network, the hypernetwork which automatically produces the relevant parameters of the image encoder based upon the particular task at hand. Specifically, the HyperCLIP hypernetwork takes as input the set of text embeddings created by the text encoder, and produces as output (some subset of) the parameters of the image encoder. The intuition here is that a suitably large hypernetwork can contain the logic of how to "specialize" the image encoder for a given task, precisely the task specified at embedding images which are assumed to be linked to one of the provided text embeddings.

At training time, all three components of the network are trained simultaneously using a contrastive (or e.g., SigLIP-based) loss. Important to emphasize, however, is the fact that at *test time*, *only* the small image encoder actually produced by the hypernetwork based upon the desired set of class prompts is used. In other words, the networks produced by HyperCLIP can be directly applied to efficient test-time classification without the need for a separate distillation phase to "shrink" the network to some smaller target architecture. This setup is illustrated in Figure 1.

### 3.2 ARCHITECTURAL DESIGN CHOICES

Several design choices informed our development of the precise architecture for the HyperCLIP model, namely in the choice of image encoder and the design of the hypernetwork itself.

**Image and text encoders.** As mentioned previously, the text encoder we use is precisely the same text encoder as in the traditional CLIP architecture, namely one based upon a causal Transformer architecture. Indeed, we could potentially use a pre-trained text encoder if desired, though we train these from scratch so as to allow for additional freedom in determining the resulting contrastive embedding space.

For the image encoder, we consider several potential small vision architectures, including EfficientNet (B0, B1, B2) (Tan & Le, 2019), MobileNetV3 (M0 and and M1) (Howard et al., 2019),TinyNet (T0) (Han et al., 2020), EdgeNext (E0) (Maaz et al., 2022), and MobileViT (V0) (Mehta & Rastegari, 2021). These have varying numbers of parameters and different architectures (see Table 1). The choice of image encoder architecture is largely dependent upon the target architecture at deployment time, and virtually any efficient vision architecture could be leveraged here.

**Hypernetwork architecture.** The HyperCLIP hypernetwork takes as input a set of text embeddings and must output the parameters of the model for the target image encoder model. This setting leads to some natural constraints and invariance that are desirable in the hypernetwork itself, as well as important considerations about what parameters are being produced.

One feature of our hypernetwork setting is the the produced network should be able to take, as input, any number of text input embedding vectors; in other words, the model should be able to produce a reasonable image encoder not just for a fixed batch size of potential prompts, but indeed for any number of prompts (up to some reasonable limit on size constraints). Additionally, the network architecture should be invariant to the ordering of these text embeddings: the "order" of the prompts provided to the hypernetwork is entirely incidental, and should have no bearing on the network being output.

Fortunately, the Transformer architecture (with variably-sized collections of inputs, and with no causal masking or position encoding) already satisfies these two desiderata. The self-attention operator used in the Transformer treats all positions in the input symmetrically. Thus, we use a non-causal Transformer architecture as the hypernetwork, with each individual prompt embedding serving as a single "token" input to the model used to produce the final parameters of the image encoder (alternatively, we can also use global average pooling over the last layer of embeddings in the hypernetwork, though in practice this causes little difference in performance). The resulting network should take *all* the inputted prompts and output a *single* set of image encoder parameters that produces an image encoder capable of maximally distinguishing between images corresponding to all such prompts. This architecture is shown in Figure 2.

**Parameters output by the hypernetwork.** The second consideration for the hypernetwork involves *which* parameters of the image encoder to actually specify as the output of the hypernetwork. While in theory it would be possible to output all the parameters of the network, in practice even relatively efficient image encoders still have millions of parameters, making it impractical to specify all of them using a hypernetwork. Instead, we adopt the approach of only modifying the BatchNorm (or LayerNorm, as appropriate) *bias* and *scale* parameters of the target image encoder; image encoders in our class of small models typically have on the order of tens of thousands of such parameters, making them a valuable target for the hypernetwork, in that they still are known to provide a very powerful control surface of the target model, while being relatively small in number. Indeed, past work has shown that it is occasionally possible to achieve reasonable performance in a deep network while *only* adjusting BatchNorm parameters (Frankle et al., 2020).

We emphasize that we *also* train the remaining parameters of the image encoder (i.e., convolution filters and MLP weights), but we do so in manner that is shared across all the different prompts within CLIP training: that is, these non-Batch/LayerNorm parameters are shared over all different batches of training, while only the Batch/LayerNorm parameters are adapted according to the output of the hypernetwork.

### 3.3 TRAINING AND LOSS

We now describe the proposed HyperCLIP training and inference steps. This setup is similar to SigLIP and other CLIP variants, with the usual image and text encoders $\mathcal{F}$ and $\mathcal{G}$, but crucially, the image encoder is *far smaller* than existing vision-language models, ideally of a size appropriate for embedded or mobile applications.

Alongside these, HyperCLIP introduces a hypernetwork $\mathcal{H}\colon \mathbb{R}^{B \times D} \to \mathbb{R}^M$ that transforms text embeddings outputted from $\mathcal{G}$ into a set of parameters of dimensionality $M$. This hypernetwork dynamically generates parameters for the image encoder $\mathcal{F}$ (or, equivalently, the image encoder acts as the mainnet for this hypernetwork). In practice, the hypernetwork's output generates only a

subset of the parameters used by the image encoder; we denote this set $\Theta'$ (so that $|\Theta'| = M$), and the remaining image encoder parameters $\Theta_{\text{fixed}}$. Specifically, in our experiments, the parameters $\Theta'$ generated by the hypernetwork are exactly the parameters inside the image encoder's normalization layers.

A forward pass through HyperCLIP with a batch of paired images and captions proceeds as follows:

$$
\begin{aligned}
Y &= \mathcal{G} \, (\text{captions}) \\
\Theta' &= \mathcal{H}(Y) \\
X &= \mathcal{F} \, (\text{images}; \Theta_{\text{fixed}} \cup \Theta')
\end{aligned}
\tag{2}
$$

With image and text embeddings $X$ and $Y$, we may calculate the SigLIP training objective $\mathcal{L}_{\text{SigLIP}}(X, Y)$. Backpropogation of this loss will allow us to train all three HyperCLIP components (image encoder, text encoder, and hypernetwork) end-to-end. Though the image encoder's parameters are partially generated dynamically by $\mathcal{H}$, its other parameters $\Theta_{\text{fixed}}$ are updated during each training iteration. During training, we freeze the normalization parameters and keep the scale parameters $\gamma$ positive by applying the exponential function, and use the running average estimate of the population statistics.

As with other vision-language models, HyperCLIP may be used as a zero-shot image classifier once trained. To do so, we follow the steps above, first generating text embeddings $Y$ from a set of class descriptions, then passing these embeddings through $\mathcal{H}$ to obtain image encoder parameters $\Theta'$. In this way, we obtain a small, deployment-friendly image classifier $\mathcal{F}$ with weights dynamically generated to match the text embeddings $Y$. Crucially, the relatively large text encoder and hypernetwork require *only one* forward pass in this process; after this, we only need the small image encoder $\mathcal{F}$ for classification.

Once we generate a task-specific zero-shot classifier, we may further finetune a linear layer (i.e., linear probe) with its weights initialized with $Y$ by minimizing the cross-entropy loss with respect to those weights: $\mathcal{L}_{\text{CE}}\left(Y^*, \text{softmax}(XY^\intercal)\right)$ where $Y^* \in \mathbb{R}^B$ are evaluation labels for each image. For zero-shot classification, each image embedding $x$ is explicitly conditioned on $Y$ using the hypernetwork before the argmax: $\arg\max\limits_{i \in \{1,\dots,K\}} Yx$.

The image embedding is obtained only after the normalization parameters of the image encoder have been modified by the hypernetwork during the forward pass. During the backward pass, the text encoder, the remaining parameters of the image encoder, and the hypernetwork are updated using the gradient of SigLIP loss computed using $Y$ and $X$.

In a typical CLIP setup, at inference time, the captions or prompts are fixed for the classes being inferred. Therefore in our setting, we can obtain the desired prompts and use them to fix the parameters of the associated image encoder before starting inference. Since HyperCLIP does not modify or add any parameters to the image encoder at inference time, the cost remains unchanged relative to a baseline model.

## 4 EXPERIMENTAL SETUP

Our main architectural contribution is to introduce an hypernetwork (hypernet) in the SigLIP architecture adapting parameters of the image encoder dynamically. Thus, training both a SigLIP model and a HyperCLIP model on the exact same image encoder, text encoder architecture, dataset for the exact same number of samples, steps, and batch size allows us to observe the marginal improvement of HyperCLIP for each of the eight image encoder architectures. The image encoder architecture is fixed that is we do not introduce any specialized modules or parameters to be adapted. The hypernet $\mathcal{H}$ is composed of an input projection layer with learnable weights $FF_{\text{input}}$, followed by a transformer encoder, a bottleneck layer, layer normalization, and an output feed-forward layer $FF_{\text{output}}$. We show the architecture in Figure 2. Across all experiments, the transformer encoder is a 12 layer transformer model, has a width of 768, 8 heads, feed forward dimension of 2560 with GELU activation, no masking, and dropout of 0.1.

The latent representation $Y$ used to compute the contrastive loss is obtained from the EOT token for each text sequence. This same representation is provided as input to the hypernet. Alternative

Table 1: We show the architecture details for each image encoder, the type and dimension of normalization layers (batch norm, layer norm, or group norm) adapted, as well as the impact on the training throughput from introducing the hypernet.

| Model | B0 | B1 | B2 | M0 | M1 | T0 | E0 | V0 |
|---|---|---|---|---|---|---|---|---|
| # Param (M) | 4.6 | 7.2 | 8.4 | 4.9 | 2.0 | 1.7 | 7.6 | 4.7 |
| # Patches | 224 | 240 | 260 | 224 | 224 | 152 | 320 | 224 |
| # Adapt (K) | 42.1 | 62.1 | 67.6 | 24.4 | 12.1 | 17.1 | 8.8 | 15.5 |
| Type Adapt | BN | BN | BN | BN | BN | BN | LN | BN&GN |
| Bottleneck dim | 285 | 193 | 177 | 491 | 577 | 512 | 256 | 512 |
| ↑ % Throughput | -3.3 | -11.2 | -18.8 | 0.16 | 0.16 | -12.7 | -47.7 | 6.8 |

representations including a class embedding did not improve performance. In the hypernet, we take the mean of the representation after the layer norm across each mini-batch before providing it as input to $FF_{\text{output}}$. The output dimension of $FF_{\text{output}}$ is the number of parameters being adapted $|\Theta|$.

We train each SigLIP model and the corresponding HyperCLIP model on 128M samples (1 epoch) of a filtered DataComp dataset (Gadre et al., 2024). See Appendix A.5 for construction of the training dataset. We evaluate on classification tasks with test sets of ImageNet-1K (IN-1K), CIFAR-100 (C100), CIFAR-10 (C10), Caltech-101 (Ca101), Food101 (F101), Oxford-IIIT Pet (Pet), Pascal VOC 2007 (VOC), STL-10. We report the **top-1 zero-shot accuracy**. We also evaluate on retrieval tasks with test sets of Flickr30k, and a subset of MSCOCO 2014. We report **top-1 mean recall**.

Finally, we evaluate on ImageNet distribution shift datasets – ImageNet-R (IN-R), and ImageNet-O (IN-O) and fairness datasets – GeoDE and Dollar Street where we report **worst-group top-1 zero-shot accuracy**. ImageNet-R comprises renditions of 200 ImageNet classes and ImageNet-O is constructed from 200 ImageNet classes with examples specifically chosen because they are misclassified with high confidence by a ResNet-50 model. Both are commonly used in out-of-distribution benchmarks. During inference, we use the same publicly available captions/prompts in OpenCLIP for both SigLIP and HyperCLIP for each dataset. See Appendix A.3 for details of evaluation datasets.

Across all experiments, we train with batch size of 1500 on four RTX A6000 GPUs with automatic mixed precision. To measure throughput, we find the maximal batch size that fits on one A100 GPU for each model without triggering an out-of-memory error or reaching Pytorch's int-32 indexing limits during the forward pass. We report the relative decrease in throughput from the additional hypernet. Note that different models vary in throughput due to specific architectural choices e.g. EdgeNext splits input tensors into multiple channel groups and this may allow for training on larger batch size without reaching the aforementioned indexing limits. However, we are only concerned with the relative impact of HyperCLIP training.

We introduce an optional learnable weight scale parameter $w_s$, which is applied to the output of $FF_{\text{output}}$. A heuristic for initializing $w_s$ involves training the SigLIP and the corresponding HyperCLIP for a few steps, measuring the norm of the parameters being adapted and the output of the hypernet, and then setting $w_s$ to a value that roughly scales the hypernet output to be similar. See Appendix A.2 for more details.

We fine-tune a linear classification head on top of the features from the image encoder (i.e. linear probing) with the train sets of ImageNet-1K and CIFAR-100 and evaluate the accuracy. This classifier is initialized with the text embedding from the text-encoder using prompts corresponding to class labels for each dataset. ImageNet-1K linear probing is trained for 10 epochs, and CIFAR-100 is trained for 100 epochs. Both optimize the cross-entropy loss using decoupled weight decay Adam with weight decay of 0.1 and learning rate of 1e-4.

We also evaluate the performance of HyperCLIP without the transformer. This effectively results in the bottleneck layer, a linear layer, being the main architectural component of the hypernet and reduces the additional training and inference latency of HyperCLIP. We present the bottleneck dimension in Table 1 for each of the models, along with the number and type of parameters adapted and the patch size for the image encoders.

## 5 RESULTS

We present the results of training HyperCLIP compared directly with the baseline SigLIP (i.e. removal of the hypernet) for each of the eight image encoder models in Table 2. HyperCLIP consistently outperforms the baseline on several of the benchmark datasets and models. Additionally, for models within the same family, such as the EfficientNet models, a HyperCLIP version of a smaller model (i.e. B0) often performs similarly to the baseline model scaled up to a larger size (i.e. B1).

Consider the zero-shot accuracy on ImageNet-1K, which is typically considered a good proxy for the performance of a CLIP model. On ImageNet-1K, we improve the performance of EfficientNetB0 by 2.4%. EfficientNetB1 is a scaled-up version of EfficientNetB0 with 2.6M additional parameters, and is only better than the HyperCLIP-adapted EfficientNetB0 by 0.3%. Similarly, HyperCLIP-adapted EfficientNetB1 outperforms EfficientNetB2 by 1% despite an additional 1.2M parameters in EfficientNetB2. On TinyNet, we improve the performance on ImageNet-1K by 3.3%.

Additionally, if we train with the optional weight scale parameter, we further improve the performance of EfficientNetB0, EfficientNetB1 and TinyNet by 0.15%, 0.68%, and 0.45% respectively on ImageNet-1K.

These results are consistent across several datasets, and we show the full table of zero-shot classification results (without weight scale parameter) in Table 2, and additional zero-shot classification results in Appendix Table 3. We also perform an ablation analysis by varying the batch size during the training of the EfficientNetB0 model. HyperCLIP consistently outperforms SigLIP, regardless of the batch size used in the analysis (see Appendix Table 4).

We also experiment with further fine-tuning zero-shot classifiers derived from both HyperCLIP and our SigLIP baseline to explore the benefits of HyperCLIP in settings where task-specific data is available. Table 2 show the results when we fine-tune a linear classification head on ImageNet-1K and CIFAR-100 data. We find that the performance of HyperCLIP's zero-shot classifier is not diminished

Table 2: Comparison of HyperCLIP and SigLIP models trained on 128M samples from the DataComp large pool using text-based and DFN filters. Metrics include Top-1 zero-shot accuracy, top-1 mean recall, and worst-group Top-1 zero-shot accuracy across various tasks. 'Arch' denotes the image encoder architecture, and HC indicates experiments using HyperCLIP.

| Model | | Classification | | Shifts | | Retrieval | | Fairness | | Fine-tuning | |
| Arch. | HC | IN-1K | C100 | IN-R | IN-O | Flickr | COCO | DS | GeoDE | IN-1K | C100 |
|-------|-----|-------|------|------|------|--------|------|------|-------|-------|------|
| B0 | ✗ | 40.2 | 53.3 | 41.0 | 55.1 | 37.6 | 22.8 | 48.5 | 72.3 | 47.6 | 65.7 |
| B0 | ✓ | **42.6** | **55.0** | **44.6** | **57.0** | **41.2** | **24.7** | **50.0** | **72.7** | **50.1** | **66.9** |
| B1 | ✗ | 42.9 | 56.6 | 44.0 | 54.3 | 41.6 | 24.9 | 48.7 | **74.9** | 50.9 | 68.1 |
| B1 | ✓ | **45.1** | **57.9** | **47.8** | **55.6** | **43.9** | **26.6** | **49.1** | 74.6 | **53.2** | **69.0** |
| B2 | ✗ | 44.1 | 56.6 | 45.3 | 56.4 | 42.8 | 25.5 | 48.7 | 75.4 | 52.5 | 68.6 |
| B2 | ✓ | **46.6** | **59.1** | **50.5** | **57.7** | **45.9** | **28.4** | **52.2** | **75.5** | **55.0** | **70.1** |
| M0 | ✗ | 29.7 | 42.3 | 28.3 | 46.1 | 25.9 | 16.0 | 43.2 | 60.8 | 35.5 | 58.1 |
| M0 | ✓ | **32.6** | **47.9** | **32.4** | **49.5** | **29.1** | **17.6** | **44.5** | **64.2** | **37.7** | **60.6** |
| M1 | ✗ | 38.3 | 49.4 | 37.5 | 52.5 | 35.8 | 21.9 | 47.4 | 68.7 | 44.9 | 62.6 |
| M1 | ✓ | **40.3** | **52.6** | **40.4** | **54.7** | **37.0** | **23.1** | 47.4 | **71.1** | **46.9** | **64.9** |
| T0 | ✗ | 29.5 | 43.1 | 29.6 | 45.3 | 26.5 | 15.8 | 42.3 | 60.3 | 35.7 | 58.0 |
| T0 | ✓ | **32.8** | **46.4** | **33.1** | **50.3** | **29.2** | **17.5** | **43.9** | **63.2** | **38.2** | **59.4** |
| E0 | ✗ | 43.5 | **56.8** | 45.3 | **57.7** | 41.1 | 25.3 | 49.1 | 74.2 | 50.4 | **68.7** |
| E0 | ✓ | **44.6** | 55.9 | **47.6** | 57.3 | **43.3** | **26.7** | **51.4** | **75.0** | **51.9** | 66.5 |
| V0 | ✗ | 36.7 | 48.7 | 35.6 | **51.2** | 33.5 | 20.1 | **48.9** | 69.7 | 45.2 | 60.5 |
| V0 | ✓ | **37.7** | **50.6** | **36.8** | 50.4 | **35.6** | **20.9** | 48.2 | **70.4** | **45.7** | **60.6** |

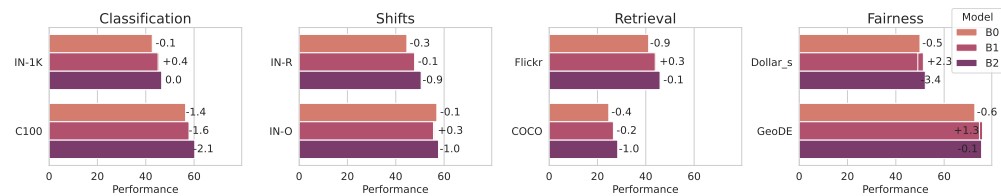

Figure 4: Performance delta when the transformer in HyperCLIP is removed on family of EfficientNet models. We report top-1 zero-shot accuracy on classification tasks, top-1 mean recall for retrieval tasks, and worst-group top-1 zero-shot accuracy for fairness tasks.

relative to the similarly fine-tuned SigLIP model, providing evidence that HyperCLIP's adaptation of the image encoder successfully incorporates additional foundation model knowledge that cannot be recovered by the SigLIP model via task-specific training.

In addition, we present the results on ImageNet-R and ImageNet-O. HyperCLIP maintains its superior performance across all the models that strictly use BatchNorm. We find that adapting LayerNorm parameters, as in EdgeNext, or combining BatchNorm and GroupNorm parameters, as in MobileViT, does not perform as well as the other models with BatchNorm interspersed in the convolution layers. Finally, we present the results on Dollar Street and GeoDE – reporting worst group top-1 accuracy. HyperCLIP outperforms the baseline SigLIP on both datasets as well.

Given that we solely focus on adapting the normalization parameters, we can provide a weak upper bound on the performance of HyperCLIP in this setting by fine-tuning the normalization parameters in addition to the linear layer parameters of the baseline Siglip model. This allows us to measure the improvement in performance compared to not fine-tuning the normalization parameters. This analysis, which we present in Figure 3, reveals how much HyperCLIP can improve a baseline SigLIP model by only adapting the normalization parameters while keeping the classification head or linear layer parameters fixed. In other words, it shows the theoretical marginal increase when adapting the normalization parameters.

On the CIFAR-100 dataset, the results show that HyperCLIP outperforms the baseline SigLIP by an average of 1.83%, while the difference between SigLIP-Probing and the SigLIP-Upper Bound averages at 3.34% across the three EfficientNet models considered.

We explore to what extent the text encoder in HyperCLIP itself aids the additional role of parameter adaptation. We present the performance delta when we remove the transformer in the hypernet. Interestingly, we find that on the EfficientNet models, HyperCLIP does not degrade significantly. For example, performance degrades by 1.7% on average on CIFAR-100. However, the individual loss in performance can be as high as 3.4% e.g. on Dollar Street with the B2 model. See Figure 4.

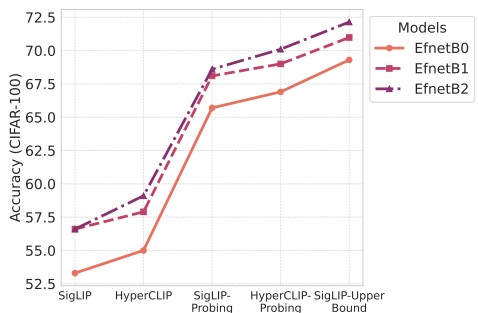

Figure 3: (Left) Impact of Normalization finetuning: HyperCLIP improves by 1.83% over SigLIP on CIFAR-100, with a 3.34% gap between SigLIP-Probing and SigLIP-Upper Bound.

HyperCLIP introduces some training throughput overhead since we need to train an additional hypernet but we find that in practice, this overhead is in-fact minimal. See Table 1. In the worst instance, the training throughput for HyperCLIP EdgeNext models is 48% worse than the baseline. However, this model uses LayerNorm – which often introduces a parallelization bottleneck due to its sequential accumulation when computing its statistics. In addition, HyperCLIP only leads to minor improvements over the baseline when LayerNorm is used. We hypothesize the expressiveness of BatchNorm, a phenomena that is well documented in the literature Frankle et al. (2020), is a crucial ingredient to its adaptability with HyperCLIP.

## 6 RELATED WORK

Most closely related to HyperCLIP, is the architecture presented by De Vries et al. (2017), which introduces Conditional Batch Normalization (CBN) for visual question answering tasks by effectively modulating BatchNorm layers of pre-trained ResNet models with linguistic input. Hypernetworks have also been used to personalize text-image models (Ruiz et al., 2023), and the notion of conditioning normalization parameters is used in image generated with GANs and diffusion models (Perez et al., 2018; Karras et al., 2019; Peebles & Xie, 2023). Our work differs in that we consider CLIP models, a more modern architecture, and consequently focuses on learning visual classifiers from natural language supervision. We focus on an end-end learning process and while the current results demonstrate the adaptablilty of BatchNorm layers, future architectures may involve the hypernet producing the entire image encoder as our knowledge of the dynamics of hypernetwork training improves. We also note the various vision-language pre-training alternatives to SigLIP such as those proposed by Jia et al. (2021); Sun et al. (2023b); Li et al. (2023); Fini et al. (2023); Lai et al. (2024); Lavoie et al. (2024). Our architectural contribution is orthogonal to the contrastive loss used and future work may explore the sensitivity to the specific contrastive loss. Finally, there exist literature toward more efficient vision-language models that distill from a larger model to a smaller model (Wu et al., 2023; Vasu et al., 2024). We emphasize that HyperCLIP skips an explicit distillation process entirely, producing an image encoder that is optimized for the specific classification task.

**Connections to Low-Rank Adaptation (LoRA).** LoRA (Hu et al., 2021) and HyperCLIP share the overarching goal of efficient model adaptation but differ significantly in their approaches and underlying mechanisms. LoRA introduces new trainable parameters during adaptation, which are added to the original model in the form of low-rank matrices. The adaptation process in LoRA typically involves two distinct steps: pre-training the base model on a large dataset, followed by fine-tuning using LoRA parameters on a specific task. In contrast, HyperCLIP does not introduce any new parameters to the classifier. Instead, HyperCLIP focuses solely on modifying the inference process without requiring additional training, thus making it a purely inference-driven adaptation method.

In addition, LoRA was primarily designed to reduce the computational and memory costs associated with fine-tuning large-scale models, such as transformer-based language models. However, the literature on using LoRA in the small-model regime or for adapting vision-language contrastive models (like CLIP) is limited. This gap highlights that LoRA's primary focus has been on tackling the challenges posed by large models, and not in resource-constrained settings.

## 7 CONCLUSION

In this paper, we introduced HyperCLIP, a new architecture designed to enhance vision-language models by dynamically adapting the image encoder using a hypernetwork. Our approach addresses the challenge of deploying large vision-language models in resource-constrained environments by producing smaller, task-specific image encoders that maintain high performance. By conditioning the image encoder parameters on the text embeddings, HyperCLIP achieves consistent and significant improvements in zero-shot accuracy, robustness to distribution shifts, and enhances fairness metrics without the need for extensive post-hoc optimization or specialized hardware.

One limitation of HyperCLIP is the potential difficulty in scaling hypernetworks which we side-step by only adapting the normalization parameters. Additionally, while HyperCLIP reduces the size of the image encoder, it introduces an overhead from the hypernetwork, which may not be negligible for extremely resource-constrained environments.

**Broader impacts.** HyperCLIP's ability to produce efficient, high-performing vision-language models has implications for democratizing computer vision models enabling their deployment on resource-limited devices and in diverse settings.

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

# A APPENDIX / SUPPLEMENTAL MATERIAL

## A.1 ADDITIONAL RESULTS ON ZERO-SHOT EVALUATION OF HYPERCLIP

Table 3: Zero-shot results on SigLIP models with various image encoding architectures. 'Arch' specifies the image encoder architecture and HC specifies the experiments in which HyperCLIP was used. Numbers reported are Top-1 zero-shot accuracy.

| Arch | HC | Ca101 | C10 | F101 | Pet | VOC | STL-1 |
|------|----|-------|-----|------|-----|-----|-------|
| B0 | ✗ | 76.2 | 80.9 | 48.9 | 57.0 | 60.3 | 88.3 |
| B0 | ✓ | **78.9** | **82.8** | **51.4** | **60.3** | **63.1** | **89.0** |
| B1 | ✗ | 78.2 | **84.4** | 52.5 | 62.5 | **65.1** | 88.7 |
| B1 | ✓ | 78.2 | 84.0 | **55.0** | **63.4** | 63.4 | **89.9** |
| B2 | ✗ | 78.5 | 84.5 | 53.8 | 63.5 | **66.4** | 90.5 |
| B2 | ✓ | **81.1** | **85.4** | **56.5** | **66.0** | 65.4 | **90.6** |
| M0 | ✗ | 67.5 | 68.3 | 37.2 | 47.5 | 52.8 | 75.2 |
| M0 | ✓ | **71.2** | **74.1** | **39.2** | **53.3** | **55.0** | **80.7** |
| M1 | ✗ | 74.2 | 78.4 | 46.4 | 56.8 | 62.5 | 86.4 |
| M1 | ✓ | **76.4** | **80.8** | **49.2** | **57.0** | **63.1** | 86.4 |
| T0 | ✗ | 69.2 | 69.1 | 35.6 | 47.9 | 42.6 | 78.1 |
| T0 | ✓ | **73.6** | **74.0** | **39.4** | **51.4** | **53.1** | **80.1** |
| E0 | ✗ | **80.1** | **84.9** | 52.6 | 60.5 | 62.9 | 90.2 |
| E0 | ✓ | 79.0 | 84.3 | **54.3** | **63.5** | **65.4** | **90.7** |
| V0 | ✗ | **71.9** | 75.4 | 47.7 | 51.4 | **59.8** | 86.1 |
| V0 | ✓ | 71.6 | **78.5** | **48.9** | **55.1** | 59.6 | **87.6** |

Table 4: Comparison of HyperCLIP zero-shot results with SigLIP models using only the Efficient-NetB0 image encoder ablating the training batch size.

| Batch size | IN-1K | C100 | IN-R | IN-O | Flickr | COCO | DS | GeoDE |
|------------|-------|------|------|------|--------|------|-----|-------|
| 500 | 36.4 | 51.3 | 38.2 | 50.9 | 34.7 | 20.3 | 48.8 | 71.6 |
| HyperCLIP | 38.7 | 53.9 | 41.4 | 53.5 | 37.6 | 21.9 | 49.1 | 72.4 |
| 700 | 38.0 | 53.1 | 39.3 | 53.6 | 35.3 | 21.7 | 49.2 | 71.9 |
| HyperCLIP | 39.7 | 53.8 | 42.9 | 53.4 | 38.3 | 22.8 | 49.5 | 72.4 |
| 1000 | 39.4 | 52.0 | 40.3 | 53.9 | 36.6 | 22.1 | 49.3 | 70.7 |
| HyperCLIP | 41.7 | 55.1 | 44.2 | 54.6 | 38.6 | 24.1 | 49.3 | 73.2 |
| 1700 | 40.4 | 53.1 | 40.8 | 54.5 | 37.6 | 22.7 | 48.6 | 69.9 |
| HyperCLIP | 42.9 | 55.6 | 44.7 | 56.5 | 41.1 | 25.3 | 50.5 | 73.9 |

## A.2 WEIGHT SCALE INITIALIZATION

We investigate the training dynamics of the CLIP loss, SigLip loss, and an extension involving a hypernetwork. Two neural network models, ModelX and ModelY, are designed with a simple architecture consisting of a linear layer followed by batch normalization. These models are trained to learn representations of synthetic data generated randomly over 100 epochs. The hypernetwork utilizes self-attention and generates dynamic parameter updates for ModelX.

The results are presented in two plots (Figure 5). We display the evolution of the parameter norms and the update norms across training epochs. The first plot shows the norms of the model parameters, highlighting how the CLIP and SigLip losses influence the magnitude of the learned representations.

The SigLip loss, with and without the hypernetwork, results in different trajectories, indicating the effect of incorporating a hypernetwork in regularizing the model's parameters. The second plot tracks the update norms, which reflect the magnitude of parameter updates during training. Incorporating the hypernetwork with SigLip loss initially shows an increase in parameter norm, which then stabilizes which may indicate some sort of regularization.

The analysis motivated the introduction of the weight scale parameters. Measuring the norm of the BatchNorm parameters during SigLIP training for an image encoder can be used as a heuristic for initializing the weight scale of HyperCLIP. To achieve this, we simply train both models for a few steps (e.g., 1M samples) and set the weight scale to match the output of the hypernet to the scale of the SigLIP BatchNorm parameters for the subsequent longer training cycle.

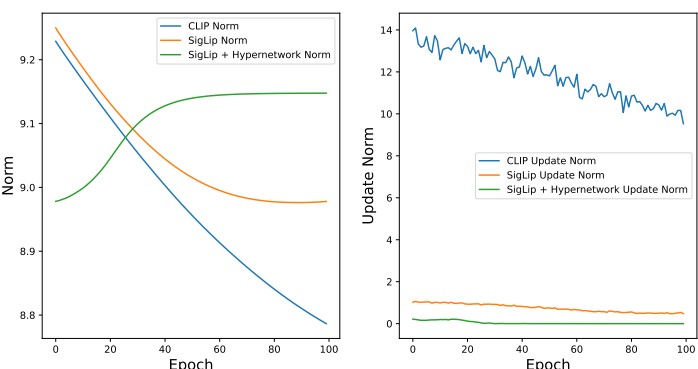

Figure 5: Evolution of parameter norms (left) and update norms (right) over 100 epochs for models trained with CLIP loss, SigLip loss, and SigLip loss with a hypernetwork. The CLIP model shows a steady decline in norms, while the SigLip models demonstrate varying behaviors, with the hypernetwork variant achieving stable updates.

## A.3 TRAINING LIBRARIES AND DATASETS

We rely on the SigLIP implementation details in the OpenCLIP (Ilharco et al., 2021) and Timm (Wightman, 2019) libraries when appropriate. The augmentation pipeline consists of randomly resizing and cropping each image, and normalizing it. We evaluate on the test sets of the following standard classification, image-text retrieval, and distribution shift and fairness benchmarks:

Evaluation prompts are gotten from publicly available CLIP benchmark: https://github.com/LAION-AI/CLIP_benchmark/tree/main/clip_benchmark/datasets

## A.4 EVALUATION METRICS

**Image retrieval recall@1** is calculated as the average number of times at least one correct image appears in the top-1 results across all captions, expressed as:

$$\text{image\_retrieval\_recall@1} = \frac{1}{N} \sum_{i=1}^{N} \left( \text{recall@1}_i > 0 \right),$$

where $N$ is the total number of captions. Similarly, **text retrieval recall@1** is calculated as the average number of times at least one correct caption appears in the top-1 results across all images, given by:

$$\text{text\_retrieval\_recall@1} = \frac{1}{M} \sum_{j=1}^{M} \left( \text{recall@1}_j > 0 \right),$$

where $M$ is the total number of images. Top-1 mean recall is the average of both measures.

| Dataset Name | Description | Number of Images / Captions |
|---|---|---|
| **ImageNet-1K (IN-1K)** | 1,000 classes of various objects. | 50,000 images |
| **CIFAR-100 (C100)** | 100 classes, with 100 images per class. | 10,000 images |
| **CIFAR-10 (C10)** | 10 classes, with 1,000 images per class. | 10,000 images |
| **Food101 (F101)** | 101 classes, with 250 images per class. | 25,250 images |
| **Oxford-IIIT Pet (Pet)** | 37 classes of pets. | 3,669 images |
| **Pascal VOC 2007 (VOC)** | 20 object classes. | 14,976 images |
| **STL-10** | 10 classes. | 8,000 images |
| **Flickr30k** | Multiple captions per image. | 1,000 images, over 5,000 captions |
| **MSCOCO 2014** | Annotated images from MSCOCO 2014. | 5,000 images |
| **ImageNet-R (IN-R)** | 200 classes focused on robustness evaluation. | 30,000 images |
| **ImageNet-O (IN-O)** | 200 classes testing out-of-distribution performance. | 2,000 images |
| **GeoDE** | 40 classes aimed at evaluating geographic diversity and fairness. | 12,488 images |
| **Dollar Street** | 58 classes from households worldwide, focusing on socioeconomic diversity and fairness. | 3,503 images |

Table 5: Overview of the datasets used in the evaluation.

## A.5 DATACOMP WITH DATA FILTERING NETWORKS

This work builds upon DataComp (Gadre et al., 2024), a benchmarking system that provides various unfiltered image-text pair pools of increasing size for evaluating CLIP models. The pools range from medium (128M datapoints) to xlarge (12.8B datapoints). Initially, we apply text-based filtering, which selects samples containing text that overlaps with ImageNet class names. Captions are identified as English using the FastText package (Bojanowski et al., 2017) and must contain words from ImageNet-21K class synsets. Next, we apply data filtering network (DFN) based filtering (Fang et al., 2023). DFNs are CLIP models trained on high-quality datasets and used to filter larger unfiltered image-text pair datasets. Specifically, the authors release the IDs of DataComp's 1.2B samples filtered with a DFN trained initially on HQITP-350M, a high-quality dataset of 357 million human-verified image-text pairs, and fine-tuned on additional datasets like MS COCO, Flickr30k, and ImageNet 1K. This filtering steps leaves us with roughly 100M unique image-text pairs.

## A.6 PROPERTIES OF SIGMOID TRAINING

The sigmoid-based loss for CLIP pre-training, termed SigLIP, is advantageous over the traditional softmax-based contrastive loss. Specifically, it is more robust to label noise, a important benefit given the inherently noisy nature of large-scale image-text datasets. SigLIP also demonstrates improved stability and efficiency across different batch sizes and performs exceptionally well at lower batch sizes, outperforming the softmax-based CLIP models with smaller batch sizes. This efficiency extends to memory usage, allowing for larger batch sizes with the same computational resources. CLIP needs to materialize a $|B| \times |B|$ matrix of pairwise similarities for a batch size $B$. For a device batch size $b$, SigLIP's "chunked" approach only requires a $b^2$ memory cost at any given moment, as opposed to the $|B|^2$ memory cost, by permuting representations across devices and computing the loss locally before summing it across all devices. While the performance gap between sigmoid and softmax losses narrows with increasing batch size, SigLIP maintains its advantages without the need for extremely large batches, which are required for optimal performance with the softmax loss.

