# OpenReview forum: "HyperCLIP: Adapting Vision-Language models with Hypernetworks"
_ICLR.cc/2025/Conference — Submitted to ICLR 2025_

### Official Review · Reviewer_qh13 · 2024-10-30

**Soundness:** 1
**Presentation:** 1
**Contribution:** 2
**Rating:** 3
**Confidence:** 4

**Summary:**

This paper proposes HyperCLIP, a vision-language architecture, as an alternative efficient solution for large vision-language models. The approach leverages hypernetworks to dynamically adjust a smaller image encoder based on input text embeddings. This approach allows large vision-language models to operate effectively with reduced resource requirements, making them suitable for edge devices and resource-constrained environments. Experiments in the paper show that the proposed approach improves SigLIP zero-shot accuracy by 3% on ImageNet and 5% on CIFAR-100.

**Strengths:**

1. The approach presents an efficient way of using smaller models to deploy vision-langauge models for resource-constrained real-world applications.
2. The adaptive approach that modifies weights at test-time for VLMs/CLIP is novel.
3. The datasets used in experiments align well with prior works.

**Weaknesses:**

1. The approach doesn't generalize to broader VLMs especially the larger models. CLIP models are generally the smaller models among recent VLMs. I don't think the proposed approach generalizes to the larger VLM models such as LLaVa [1] that uses very large decoder-only transformer LLM as component.
2. CLIP models have wide range of applications, among which an important one is to use the visual features produced by the image encoder as inputs to downstream models. HyperCLIP reduces CLIP applications to only image classification.
3. The experiments are not solid. The paper claims that HyperCLIP performs well on ImageNet in zero-shot setting. However, In Appendix A.5, the paper writes 'selects samples containing text that overlaps with ImageNet class names', which shows that the ImageNet class names are used in data collection which is used in training.  - so it's not truly zero-shot.


[1] Liu, Haotian, et al. "Visual instruction tuning." Advances in neural information processing systems 36 (2024).

**Questions:**

Can you explain why do you think your experiments are conducted in zero-shot setting?

My biggest concern is weakness #3.

---

> ### Author Response · Authors · 2024-11-20
>
> > **Can you explain why do you think your experiments are conducted in zero-shot setting?**
>
> This is certainly a fair point, but we really want to emphasize that this is exactly the terminology used by existing benchmarks and CLIP training pipelines in general. Universally, CLIP training datasets use the ImageNet (training) data in their collection of images and often filter based upon these keywords. Indeed, the setup we use here is not from our own model, but exactly follows the methodology of the DataComp paper [1], in order to maximally fit within standardized training setups. In this setup, a large pool of data in the form of images and captions is collected (often from the web), and the models are trained on this pool. Such data pools may be filtered to improve the downstream performance of the trained models.
>
> The evaluation is ultimately done on test sets of standard image-classification datasets, e.g., ImageNet, CIFAR100, etc., that are not included in the training dataset.
>
> Specifically, the filtering process in our paper involves using the ImageNet-21k class names as a filter of the captions (again introduced as a baseline in [1]). Indeed, the filter does not involve using any of the ImageNet images, and using just the words from ImageNet-21k to filter the captions is a very weak form of bias.
>
> More importantly, the *aim* of this paper is to contrast HyperCLIP and CLIP under the *exact same* setup (i.e., both zero-shot and fine-tuning) for (classification, image retrieval) and on diverse datasets (including fairness benchmarks).
>
> While the filtering process is important, it doesn't give our method any systematic advantage, and we evaluate on other datasets besides ImageNet. There is ongoing research into the extent (if any) of data leakage that might be included in these data pools, i.e., if the test sets of standard image datasets have found their way into large training data pools, but that is outside the scope of our work and, as mentioned, does not invalidate our architectural contribution.
>
> [1] DataComp: In search of the next generation of multimodal datasets NeurIPS 2023.
>
> ---
> > **The approach doesn't generalize to broader VLMs, especially the larger models. CLIP models are generally the smaller models among recent VLMs. I don't think the proposed approach generalizes to the larger VLM models such as LLaVa.**
>
> This is really a separate point, because broader VLMs and CLIP models are quite different in their nature: broader VLMs integrate vision tokens in LLMs to let the system produce text pertaining to images, while CLIP is fundamentally an image/text encoder that produces well-aligned representations. Perhaps most notably, VLMs like LLaVA *use* CLIP as their underlying tokenization of the images and benefit hugely from these pre-trained representations. Thus, the domains are really orthogonal (but also nicely related), and we suggest not discounting work on CLIP-based methods simply because they are a “small” component of larger VLMs.
>
> ---
>
> > **HyperCLIP reduces CLIP applications to only image classification.**
>
> This is not entirely correct. We also evaluate HyperCLIP on retrieval tasks, which is perhaps closer to another more popular use of CLIP models. In this setting, we measure the recall when the text embedding (of the CLIP text encoder) is used to retrieve an image from a set of images, and similarly when the image embedding (of the CLIP image encoder) is used to retrieve a caption from a set of captions.
>
> But more broadly, this approach is fundamentally about producing a highly compact, text-specific image encoder in CLIP, which certainly could find application to other domains like VLMs, etc. (as mentioned above, CLIP is the image encoder used by many open-source VLMs). While we believe this to be orthogonal to the current paper and likely a question for future work, we absolutely feel that improvements to CLIP such as this are valuable contributions to the field as a whole, even if CLIP zero/few-shot image classification is only one potential application.

---

### Official Review · Reviewer_SCQD · 2024-10-31

**Soundness:** 3
**Presentation:** 3
**Contribution:** 3
**Rating:** 6
**Confidence:** 3

**Summary:**

The paper introduces HyperCLIP, a novel architecture that adapts vision-language models by using a hypernetwork to dynamically adjust the weights of a small image encoder for each new set of text inputs.
This approach enables the creation of zero-shot deployment-friendly image classifiers with a single forward pass through the text encoder and hypernetwork.
HyperCLIP is designed to overcome the challenges of deploying large models in constrained environments by offering a smaller, yet powerful alternative.
The model increases zero-shot accuracy on ImageNet and CIFAR-100 with minimal training throughput overhead.

**Strengths:**

1. The paper proposes using a hypernetwork to generate weights for a smaller image encoder within the SigLIP contrastive pre-training framework, allowing for task-specific specialization without extensive retraining.
2. HyperCLIP achieves significant improvements in zero-shot accuracy on ImageNet and CIFAR-100, with minimal overhead, making it suitable for resource-constrained environments.
3. The method is compatible with any type of contrastive pre-training, enhancing its versatility.
4. The paper demonstrates that HyperCLIP can improve the performance of small vision models on standard benchmarks by adapting only the normalization layers.
5. HyperCLIP has the potential to democratize computer vision by enabling the deployment of high-performing models on devices with limited resources.

**Weaknesses:**

1. By focusing on adapting only normalization parameters, the paper may not fully leverage the potential of hypernetworks to modify other model parameters.

**Questions:**

1. Will you plan to release the code ?

---

> ### Author Response · Authors · 2024-11-20
>
> > Will you plan to release the code ?
>
> Yes, we will publish the code and model weights after the review process.

---

### Official Review · Reviewer_GRgs · 2024-11-04

**Soundness:** 2
**Presentation:** 1
**Contribution:** 1
**Rating:** 3
**Confidence:** 5

**Summary:**

This paper proposed a hyper-network to generate parameters for normalization layers in the vision model of CLIP. The paper is aimed at solving the efficiency problem of CLIP on edge computing scenarios. The experiment results show that the proposed method can get better zero-shot accuracy over the CLIP baseline.

**Strengths:**

1. As shown in Table 2, the proposed hyper-network can improve the zero-shot accuracy on ImageNet by up to 3%.

**Weaknesses:**

**Unclear Motivation and Unsupported Claims by Experiment Results**

The motivation for using a hyper-network to address CLIP's efficiency challenges in edge computing applications is unclear. The introduction does not clearly explain this choice. The experiments show performance improvement over a baseline without a hyper-network but fail to address the efficiency problem. In other words, the hyper-network enhances performance only in small-scale models rather than improving CLIP's efficiency directly.

**Poor Writing and Difficult to Follow**

* Several English expressions are incorrect, likely due to machine translation. For instance, in lines 041-044: "These methods often include first training a large model, and then applying the chosen technique in a post-hoc fashion. Additionally, many of these methods can require specialized hardware support for actual memory and latency reduction." This section is hard to understand.

* The text includes many excessively long sentences. For example, in lines 045-050: "We propose a method of pre-training vision-language models (VLMs) that allows us to derive small vision models appropriate for deployment on edge devices without requiring multi-step training procedures or any specialized hardware. We suggest a new contrastive learning architectural design based on hypernetworks that improves performance over current state-of-the-art baselines and can additionally be used in conjunction with a variety of model compression methods for further memory or latency improvements."

**Limited References and Literature Review**

* The introduction has limited references, mentioning only three prior works (Sun et al., 2023a; Dettmers et al., 2022; Frantar & Alistarh, 2023). Including more background citations would strengthen the paper’s credibility.
* The related work section is limited to a single paragraph (L488-503). A more thorough literature review, including discussions on established methods like LoRA [a] and adaLN [b], would enhance this section and contextualize the paper’s contributions.

[a] Edward et al., "LoRA: Low-Rank Adaptation of Large Language Models." ICLR 2022.

[b] William et al., "Scalable Diffusion Models with Transformers." ICCV 2023.

**Questions:**

1. Why the text transformer (a width of 768, 8 heads, feed-forward dimension of 2560) is not a small model, since the proposed HyperCLIP is targeted for compute-restricted scenarios.
2. Is the HyperCLIP trained from scratch or initialized from SigLIP?
3. Which part of the experiments are zero-shot results, and which part are linear probing results?

---

> ### Author Response · Authors · 2024-11-20
>
> We thank the reviewer for their thorough review.
>
> ### Re: Questions
>
> > **Why the text transformer (a width of 768, 8 heads, feed-forward dimension of 2560) is not a small model, since the proposed HyperCLIP is targeted for compute-restricted scenarios.**
>
> As discussed in the paper, the cost of inference when deploying a CLIP model, e.g., for classification, predominantly comes from the image encoder. This is because we pre-specify the prompts for each class and then obtain the corresponding text embedding using the text encoder. During inference, we don’t need another pass through the text encoder; however, we always require a forward pass through the image encoder. Consequently, the main bottleneck with respect to size (for inference) is the image encoder, not the text encoder.
>
> > **Is the HyperCLIP trained from scratch or initialized from SigLIP?**
>
> HyperCLIP is trained from scratch in all our experiments, as mentioned in the paper (line 215).
>
> > **Which part of the experiments are zero-shot results, and which part are linear probing results?**
>
> The last two columns of Table 2 refer to the results of fine-tuning a linear layer (i.e., linear probing) on ImageNet-1K and Cifar100. The remaining results in Table 2, as well as additional results in Appendix A1, refer to zero-shot performance.
>
> ---
>
> ### Re: Weakness
>
> > **The experiments show performance improvement over a baseline without a hyper-network but fail to address the efficiency problem. In other words, the hyper-network enhances performance only in small-scale models rather than improving CLIP's efficiency directly.**
>
> We believe there may be a misunderstanding here. We are explicitly developing a model to improve the test-time deployment of a zero-shot classifier, which directly corresponds to using a smaller image encoder.
>
> Our notion of efficiency is the model size (number of active parameters during inference). In practice, model size is the dominant factor when deploying models in edge applications, as it directly impacts latency and is often constrained by memory. This means that if we can reduce the size of a CLIP model (in particular, the image encoder) while maintaining its performance, we achieve a more efficient CLIP model.
>
> In this paper, we show that we can achieve this goal, and our experiments demonstrate that HyperCLIP models with *less* parameters perform better than CLIP. For example, HyperCLIP-adapted EfficientNetB1 outperforms EfficientNetB2 by 1% (on ImageNet-1K), despite an additional 1.2M parameters in EfficientNetB2.
>
> While we acknowledge that model size is an imperfect notion of efficiency, EfficientNet *is* explicitly an architecture designed for inference efficiency, and thus seems a reasonable choice when focusing on test-time performance.
>
> > **Poor Writing and Difficult to Follow**
>
> Thank you for highlighting specific areas for improvement. We will revise these passages for clarity, remove ambiguities, and simplify complex sentences.
>
> > **Limited References and Literature Review**
>
> Methods like LoRA are orthogonal to our work, as they focus on introducing new parameters that are then fine-tuned for specific capabilities. We will include a discussion of these methods in a revision. Could the reviewer clarify why diffusion-transformers, i.e., [b], are relevant to our paper, which focuses on the efficiency of CLIP models in resource-constrained deployment?

---

> ### Author Response · Authors · 2024-11-23
>
> We have uploaded a revised version of the paper, highlighting edits made to improve the readability of the introduction and to include additional references. Additionally, we have expanded the discussion in the related work section to address the connections between HyperCLIP and LoRA. We hope these revisions address your feedback, and we would be happy to discuss any remaining concerns.

---

### Meta-Review · Area_Chair_T7sW · 2024-12-05

**Metareview:**

### **Summary**
The paper proposes **HyperCLIP**, a vision-language architecture designed for efficient deployment of CLIP-based models in resource-constrained environments. The key innovation lies in using a hypernetwork to dynamically adapt the weights of a smaller image encoder based on text inputs. HyperCLIP achieves competitive improvements in zero-shot image classification accuracy on ImageNet and CIFAR-100, demonstrating its potential for edge computing scenarios. The paper also provides evaluations on retrieval tasks, highlighting broader applicability beyond image classification.

### **Strengths**
1. **Innovative Use of Hypernetworks**:
   - The approach demonstrates a novel application of hypernetworks for adapting normalization layers in CLIP’s image encoder, enabling task-specific specialization.
2. **Practical Relevance**:
   - HyperCLIP addresses a pressing need to deploy high-performing models on resource-constrained devices while maintaining competitive accuracy.
3. **Empirical Performance**:
   - HyperCLIP achieves up to a 3% improvement in zero-shot accuracy on ImageNet and 5% on CIFAR-100 compared to SigLIP.
4. **Compatibility and Versatility**:
   - The method integrates seamlessly with existing contrastive learning frameworks, offering a practical pathway to improve edge deployment without significant additional overhead.

### **Weaknesses**
1. **Evaluation Concerns**:
   - The paper’s claim of a zero-shot setting is undermined by the use of ImageNet-21k class names for filtering training data. While common in CLIP-based benchmarks, this setup introduces potential bias and reduces the perceived novelty of the approach.
   - Generalization to larger vision-language models (e.g., LLaVA) or tasks beyond image classification is not explored, limiting broader applicability.
2. **Presentation and Writing**:
   - Reviewers noted unclear motivations, overly complex sentences, and insufficient clarity in experimental setups. The paper would benefit from significant editing for readability.
3. **Limited Experimental Scope**:
   - Ablation studies on the choice of components (e.g., hypernetwork architecture or the focus on normalization layers) are absent.
   - The related work section is sparse, missing connections to well-established methods like LoRA and deeper discussions on compression techniques.
4. **Narrow Applications**:
   - The method primarily focuses on zero-shot image classification and retrieval tasks, narrowing its potential impact within the broader domain of vision-language applications.

### **The reason and decision**
While the proposed approach is novel and promising, it falls short in terms of evaluation rigor, clarity, and generalizability. The issues surrounding the zero-shot claims, combined with limited exploration of broader applications, prevent it from meeting the standard for acceptance at ICLR.

**Additional Comments On Reviewer Discussion:**

The authors addressed several key concerns raised during the review process:
- **Improved Clarity**: Revisions improved the introduction and related work sections, adding discussions of LoRA and additional references.
- **Detailed Responses**: Authors clarified the rationale behind focusing on normalization layers, the filtering process using ImageNet-21k class names, and the scope of the experiments.
- **Expanded Experiments**: Retrieval task evaluations were highlighted, though broader applications and ablations were not added.

While the rebuttal addressed specific points, critical gaps remain:
- **Zero-Shot Claims**: The use of ImageNet class names in filtering undermines the claim of a truly zero-shot setup, which is a significant methodological shortcoming.
- **Broader Impact**: The approach does not generalize well to larger vision-language models or tasks outside classification and retrieval, limiting its scope.
- **Writing**: While improved, the writing remains challenging to follow in parts, particularly for a general audience.

---

### Decision · Program_Chairs · 2025-01-22

Reject